# Aspects of Reproductive Biology of the European Hake (*Merluccius merluccius*) in the Northern and Central Adriatic Sea (GSA 17-Central Mediterranean Sea)

**Michela Candelma** [1], **Luca Marisaldi** [1], **Daniela Bertotto** [2], **Giuseppe Radaelli** [2], **Giorgia Gioacchini** [1], **Alberto Santojanni** [3], **Sabrina Colella** [3] and **Oliana Carnevali** [1,*]

[1]    Laboratory of Developmental and Reproductive Biology, DiSVA, Università Politecnica delle Marche, 60131 Ancona, Italy; m.candelma@univpm.it (M.C.); l.marisaldi@univpm.it (L.M.); giorgia.gioacchini@univpm.it (G.G.)
[2]    Department of Comparative Biomedicine and Food Science, University of Padua, 35122 Padova, Italy; daniela.bertotto@unipd.it (D.B.); giuseppe.radaelli@unipd.it (G.R.)
[3]    National Research Council (CNR), Institute of Biological Resources and Marine Biotechnologies (IRBIM), 60125 Ancona, Italy; alberto.santojanni@cnr.it (A.S.); sabrina.colella@cnr.it (S.C.)
*    Correspondence: o.carnevali@staff.univpm.it; Tel.: +39-0712-204-990

**Abstract:** The study focused on the macroscopic, histological, and biometric analysis of European hake females in GSA 17 (Central-North Adriatic Sea). From 2013 to 2015, 976 females were collected and analyzed monthly. Though females in spawning conditions were found during the whole year, the trend of GSI showed a peak of the reproductive season from April to July in 2014 and 2015. HSI and $K_n$ reached the highest values in September, after the spawning peaks. In 2013, the trend of these indices did not highlight an evident peak, probably due to an adverse event that occurred in the previous winter in the Adriatic shelf. The length at first maturity ($L_{50}$) was estimated by macroscopic and histological approaches, resulting in 30.81 cm for the macroscopical length and 33.73 cm for the histological length; both values are higher than the current catching legal size. For the first time in this area, batch and relative fecundity were estimated. Relative fecundity was similar to the Mediterranean and the Atlantic stocks, whereas batch fecundity values were lower compared to other fishing grounds. Overall, the analysis of reproductive parameters plays a fundamental role in the sustainable management of this resource in an area as overfished as the Central-North Adriatic Sea.

**Keywords:** European hake; *Merluccius merluccius*; fecundity; somatic indices; Adriatic Sea; $L_{50}$

## 1. Introduction

In the last several decades, the excessive fishing effort, together with the increase of pollution, poor fishing management, and impairment of marine ecosystems, caused the depletion of fish stocks worldwide. The Food and Agriculture Organization reported that 57.4% of fish stocks are fully exploited, 29.9% are overexploited, and only 12.7% are not fully exploited [1]. In the Mediterranean Sea, the long-lasting, intense fishing pressure applied on fish and invertebrate stocks has led to declining population biomasses [2], which has also been reflected in a reduction of catches for the majority of stocks [3]. Mullon et al. [4], by analyzing the FAO dataset of world fisheries catches for the period 1950–2000, detected that the major collapse occurred for demersal species. Colloca and coworkers [5] assessed the impacts of the fishing pressure in the period 2002–2014 in the Mediterranean Sea and determined that the Central-North Adriatic represents the highest catching area of demersal species in the Mediterranean area.

Among demersal species, the European hake (*Merluccius merluccius*, L. 1758) received attention because it represents one of the principal fishing targets in the Northeast Atlantic Ocean and the Mediterranean Sea. In the Mediterranean Sea, a total landing of 22,547 tons was recorded in 2011 [6,7]; in the Central-North Adriatic Sea, the European hake is one

of the leading commercial species [8] and represents 77% of landings from Croatia [9]. To avoid the collapse of hake stock, fishing should be conducted more sustainably by maintaining a spawning stock biomass level that is suitable to guarantee the renewal of the species. In this regard, egg-producing females contribute more to embryonic development success. The acquisition of information on reproductive cycle of the female Adriatic stock, including length at first maturity, fecundity, and spawning cycle, is essential to quantify the reproductive potential of this population [10]. Because of the great commercial importance of the European hake and the critical status of its stocks, several studies were performed in different areas of the Mediterranean Sea related to stock assessment, spawning cycle and fecundity estimation [11–16], feeding habits [17,18], analysis of lipid reserves [19], nursery area [6,20], juvenile recruitment [21], the selectivity of fishing gear [22], fisheries management [2,23,24], and reproductive physiology [25,26]. However, none investigated the fecundity, the spawning cycle, and the proportion of mature individuals at length by a macroscopic and histological approach in the Central-North Adriatic Sea (FAO Geographical Sub-Area 17, according to GFCM division), where this species represents a valuable economic resource.

In this regard, a multiannual survey (2013–2015) was conducted to improve understanding of the reproductive dynamics of European hake in this area. In particular, the work focused on females since the female conditions limit the egg production and the progeny production more than males, and it evaluated the size at first maturity in females using the histological and macroscopic analysis of the gonads. The condition indices such as Le Cren's condition factor ($K_n$), the hepatosomatic index (HSI), and gonadosomatic index (GSI) were calculated and compared across different ovarian stages, and their changes analyzed in the three years. Furthermore, batch and relative fecundities were estimated together with the analysis of ovarian stages.

This study has never been carried out in this area (GSA 17) and completes the scenario of this important resource of the Italian seas, in order to provide knowledge for a sustainable management.

## 2. Materials and Methods

### 2.1. Sampling

A total of 976 females ranging from 13 to 64 cm in total length (TL) were collected aboard commercial bottom trawler fishing vessels in the Northern and Central Adriatic Sea (FAO Geographical Sub-Area 17, according to GFCM division) monthly from January 2013 to December 2015. Guidelines of the Data Collection Framework Regulation (EU Reg.199/2008) followed those established by the Community system for the conservation and sustainable exploitation of fisheries resources under the Common Fisheries Policy (CFP). The sampling procedures did not include any animal experimentation and animal ethics approval was therefore not necessary under the Italian legislation (D.L. 4 March 2014, n. 26, art. 2). For each specimen, the following parameters were recorded: total weight (TW), gutted weight (GW), total length (TL), sex, macroscopic maturity stage of gonads, and gonad and liver weights.

### 2.2. Reproductive Seasonality and Fish Condition Indices

To evaluate the temporal variation of maturity and condition stage of females, the gonadosomatic index (GSI), hepatosomatic index (HSI), and Le Cren's relative condition factor ($K_n$) were calculated. These indices were defined by the following equations:

$$\text{GSI (\%) = gonad weight/gutted weight body*100}$$

$$\text{HSI (\%) = liver weight/gutted weight body*10}$$

$$K_n = W/a \, TL^b$$

where *a* and *b* are the regression parameters of the length–weight relationship, W is gutted weight, and TL is the total length. GSI was evaluated only for the mature females;

313 mature animals were found from developing to regenerating phase. From January to March of 2013, livers were not available for sampling problems. The spawning season was also investigated by analyzing the monthly frequency of ovarian maturity stages. Number of samples of mature females used for GSI, HSI and Kn calculation for each month in three years are reported in Table S1.

### 2.3. Histological Analysis

173 female specimens at different maturity stages were randomly chosen for histological analysis while choosing a group that was representative for all sizes to confirm the macroscopic classification of ovarian development. Ovarian samples were fixed at 4 °C overnight in formalin at 10% in phosphate-buffered saline (PBS, 0.1 M, pH 7.4), after which they were washed in PBS and stored in 70% ethanol at 4 °C until use. The samples were then dehydrated through a series of graded ethanol, cleared in xylene, and embedded in paraffin wax. Five μm thick consecutive sections were cut using a microtome RM2125 RTS (Leica Biosystems, Wetzlar, Germany) and stained with Meyers's hematoxylin and eosin. Images were acquired using a Zeiss Axio Imager M2 microscope (Zeiss, Oberkochen, Germany). Assessment of the reproductive status of samples was performed according to the method of Brown-Peterson and coworkers [27] but adapted for European hake following the method of Candelma and coworkers [25]. Five ovarian stages were defined: (i) immature; (ii) developing; (iii) spawning capable and actively spawning; (iv) regressing (postspawning); (v) regenerating (spent) (Table 1).

**Table 1.** Criteria used to determine the maturational status of European hake females.

| Ovarian Phases | Macroscopic Morphology | Histological Aspect | Follicle Diameter (μm) |
|---|---|---|---|
| Immature | Orange, semi-transparent | Presence of oogonia (O), primary oocytes (PO). Scarce connective tissue and well compact ovigerous lamellae. | <250 |
| Developing | Small pink but some oocytes visible | Presence of early (vtg1) and middle (vtg2) vitellogenic oocytes. Yolk vesicles form in the ooplasm. Lipid granules occupy a larger cytoplasmic area than yolk droplets. The zona radiata thickens. | 250–550 |
| Spawning capable and actively spawning subphase | Large ovaries, oocytes visible macroscopically. An abundance of hydrated oocytes in actively spawning subphase. | Presence of late vitellogenic oocytes (vtg3). Lipid globules occupy a cytoplasmic area like that occupied by yolk granules. Actively spawning subphase: lipid globules fuse into a single larger oil droplet, yolk droplets start to coalescence, the nucleus starts to migrate peripherally to the animal pole (Mn) and POFs. Follicles grow due to water uptake and become transparent (H). Zona radiata is thinner than in previous stages | 550–1150 |
| Regressing (postspawning) | Flaccid and small ovaries, blood vessels prominent. | Atresia (any stage) and POFs present. Some CA and/or vtg1, vtg2 oocytes present. | <250 |
| Regenerating | Pinkish and small ovaries that occupy 1/3 of the body cavity, blood vessels reduced but present. | Presence of oogonia (O), primary oocytes (PO), cortical alveoli (CA), and lipid stage oocytes (LS). The ovarian wall is thickened. | <250 |

The measurement of oocytes was only from oocytes that had been sectioned through the nucleus. The oocyte classification followed Candelma et al. [26] and Murua et al. [28]: oogonia (O); primary oocyte (PO); cortical alveoli (CA); lipid stage (LS); early vitellogenesis (vtg 1); middle vitellogenesis (vtg 2); late vitellogenesis (vtg 3); migrating nucleus (Mn); hydration (H); postovulatory follicles (POFs). Cohen's *k* coefficient [29] was applied to assess the agreement between the histological and macroscopic classification.

*2.4. Size at First Maturity (L$_{50}$) and Fecundity*

The proportion of maturity at length (PL) was estimated using both histological and macroscopic data with the following logistic function:

$$PL = \frac{1}{1 + \exp(\alpha + \beta * TL)}$$

where $\alpha$ (intercept) and $\beta$ (slope) represent the estimated parameters and *TL* is the total length. All specimens were used for macroscopic investigation; 173 individuals were examined using histological analysis. Females from the developing stage onwards were considered mature. The length at which 50% of the females are mature was computed as L$_{50}$ = $-\alpha/\beta$. The two regression logistic curves of histological and macroscopic data were compared with the likelihood ratio test.

With regards to batch and relative fecundities, 28 females at the actively spawning stage representing the total number of females available were used and the gravimetric method, based on the relationship between ovary weight and oocyte density as described by Murua et al. [30], was applied. Because the distribution of hydrated oocytes does not statistically vary in the ovaries [31], only a lobe of the hydrated ovary was used for analysis. The hydrated oocytes from fresh subsamples were manually counted under a stereomicroscope (Optika, Italy). Batch fecundity was estimated as the average of the hydrated oocyte number integrated over three subsamples multiplied by ovary mass for each specimen with the hydrated oocyte [11]. The relative fecundity was calculated as a ratio of batch fecundity and gutted weight for each fish.

*2.5. Statistical Analysis*

Statistical differences of the monthly variation of indices for each year and of indices per ovarian stages in each year were determined using a Tukey's multiple test comparison. Both analyses were performed using Prism 6 (GraphPad Software, San Diego, CA, USA). For the determination of size at first maturity, the R statistical environment [32] and the packages "Fisheries Stock Assessment" (FSA) [33], ggplot2 [34], and rel [35] were used. The *p*-values < 0.05 were considered as significant. Results were expressed as the mean ± SEM.

## 3. Results

*3.1. Reproductive Seasonality and Fish Condition*

The percentages of European hake females in different maturity stages observed monthly in the three years indicated that specimens from developing to regressing phase were present throughout the year (Figure 1).

In Figure 2, the GSI trend of mature females during three years of sampling is represented. The GSI values were lowest in 2013 compared to the other two years. In 2013, the highest value was recorded in April with a significant difference only compared to October (Tukey's multiple comparison test; *p* < 0.05) (Figure 2a). Differently from 2013, in 2014 and 2015, the highest peak was observed in June and the lowest in September (Figure 2b,c). In particular, in 2014, the GSI in June was significantly higher than February, September, October, November, and December (Tukey's multiple comparison test; *p* < 0.05) (Figure 2b), and the peak in July was significantly higher than September, December, and February (Tukey's multiple comparison test; *p* < 0.05). In 2015, June was significantly different only compared to September (Tukey's multiple comparison test; *p* < 0.05) (Figure 2c).

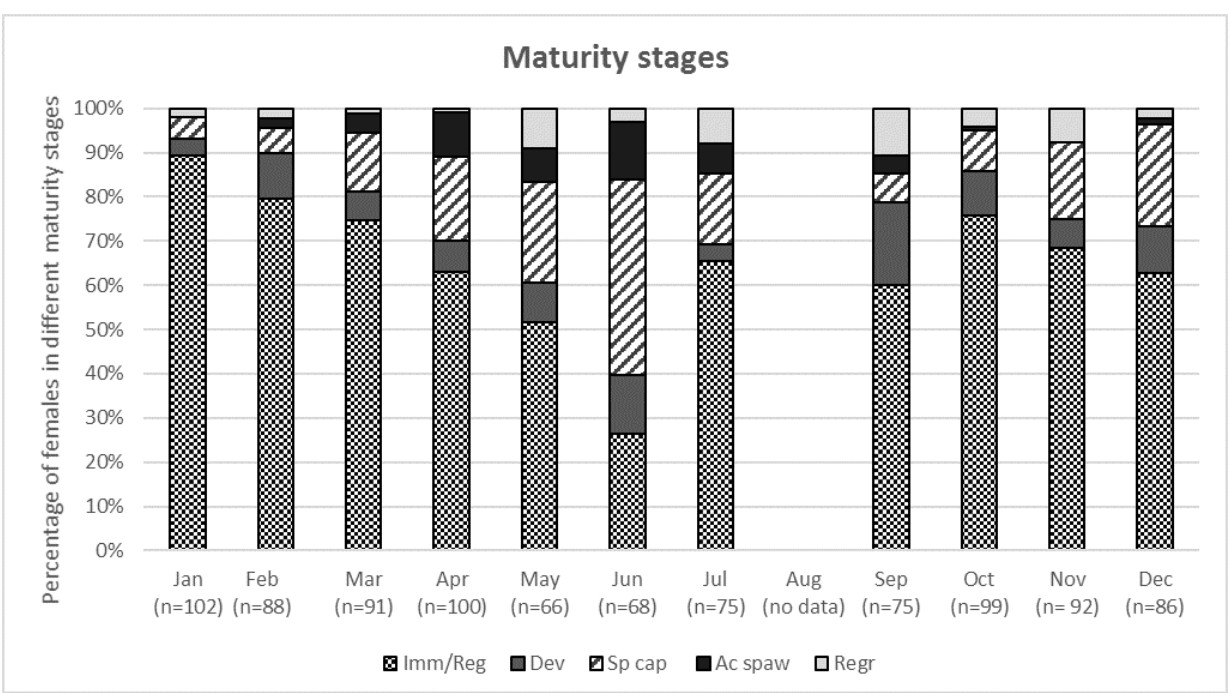

**Figure 1.** Monthly percentages of European hake female maturity stages. Imm/reg, includes individuals in immature and regenerating stages; dev, developing; spaw, spawning capable; ac spaw, actively spawning, regr, regressing. For their histological characteristic, the immature and regenerating stages were grouped (See Section 3.2).

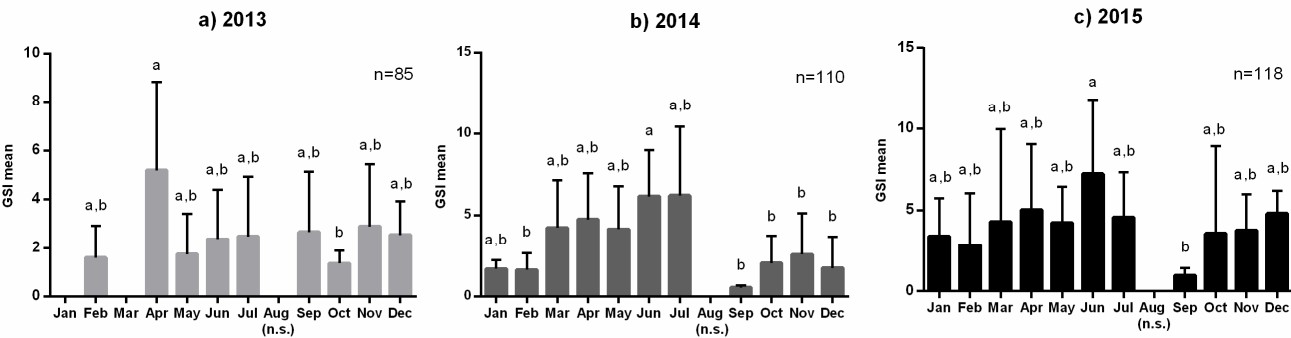

**Figure 2.** Monthly variation of European hake female gonadosomatic index (GSI) for three years. Statistical differences were described in the result section. In January and March of 2013, no mature females were found. In August of the three years, no samples (n.s.) were sampled. The letters indicate a statistically significant difference among groups ($p < 0.05$) determined by Tukey's multiple comparison test.

The HSI analysis distinguished immature females from mature females (considering from developing to regenerating phase) (Figure 3). The HSI trend varied over the three years. In immature females from 2013, the highest values were in summer months, but significantly different only with September (Tukey's multiple comparison test; $p < 0.05$) (Figure 3a). In 2014, the highest peak was reached in December, significantly different compared to March (Tukey's multiple comparison test; $p < 0.05$) (Figure 3b), whereas in 2015, the highest value was in September and significantly different only compared to March (Tukey's multiple comparison test; $p < 0.05$) (Figure 3c). The lowest value was in March in both 2014 and 2015. In 2013, as in immature females, the mature females were also characterized by an increase of HSI values from May to July (Figure 3d), whereas the lowest measurement was in April significantly diverse compared to July, September, October, November, and December (Tukey's multiple comparison test; $p < 0.05$). Different peaks were recorded during 2014, the highest in December, significantly higher than March–

July (Tukey's multiple comparison test; $p < 0.05$) (Figure 3e). In 2015, the HSI surged in September, significantly different compared to all months (Tukey's multiple comparison test; $p < 0.05$), except to January and December (Figure 3f), and the lowest value was in May.

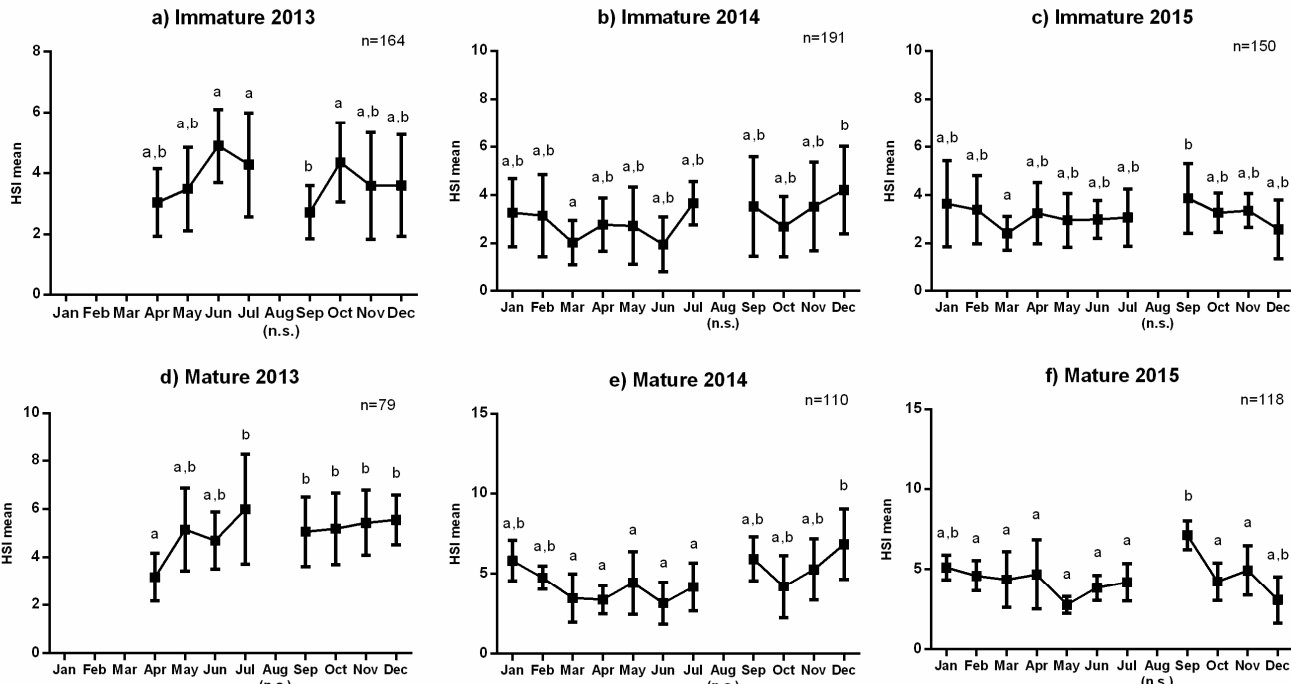

**Figure 3.** Monthly variation of European hake female hepatosomatic index (HSI) for three years in different ovarian stages. Graphics (**a**–**c**) indicate the females in the immature phase, while graphics (**d**–**f**) show females from developing to regenerating phase. From January to March of 2013, no sampling of the liver was conducted. In August of the three years no samples (n.s.) were sampled. The letters indicate a statistically significant difference among groups ($p < 0.05$) determined by Tukey's multiple comparison test.

The $K_n$ analysis distinguished immature females from mature females (considering from developing to regenerating phase) (Figure 4). In immature individuals collected in 2013, the highest peak was in October and significantly different only compared to April (Tukey's multiple comparison test; $p < 0.05$) (Figure 4a), while in mature animals no significant variation was observed (Figure 4d). The trend of $K_n$ varied mainly in 2014 and 2015, with the highest peak recorded in September for both immature and mature females (Figure 4b–f). In immature individuals captured in 2014, the lowest value was in March and significantly different compared to September and November (Tukey's multiple comparison test; $p < 0.05$) (Figure 4b), whereas mature females showed the lowest value in June with significant differences respect to September and December (Tukey's multiple comparison test; $p < 0.05$) (Figure 4e). In 2015, the $K_n$ of immature females was significantly higher in September compared to the rest of the year except for May, June, July, and October (Tukey's multiple comparison test; $p < 0.05$) (Figure 4c). In mature females captured in 2015, the highest peak of September was significantly different compared to January, April, and May (Tukey's multiple comparison test; $p < 0.05$) (Figure 4f), whereas the value was lowest in December.

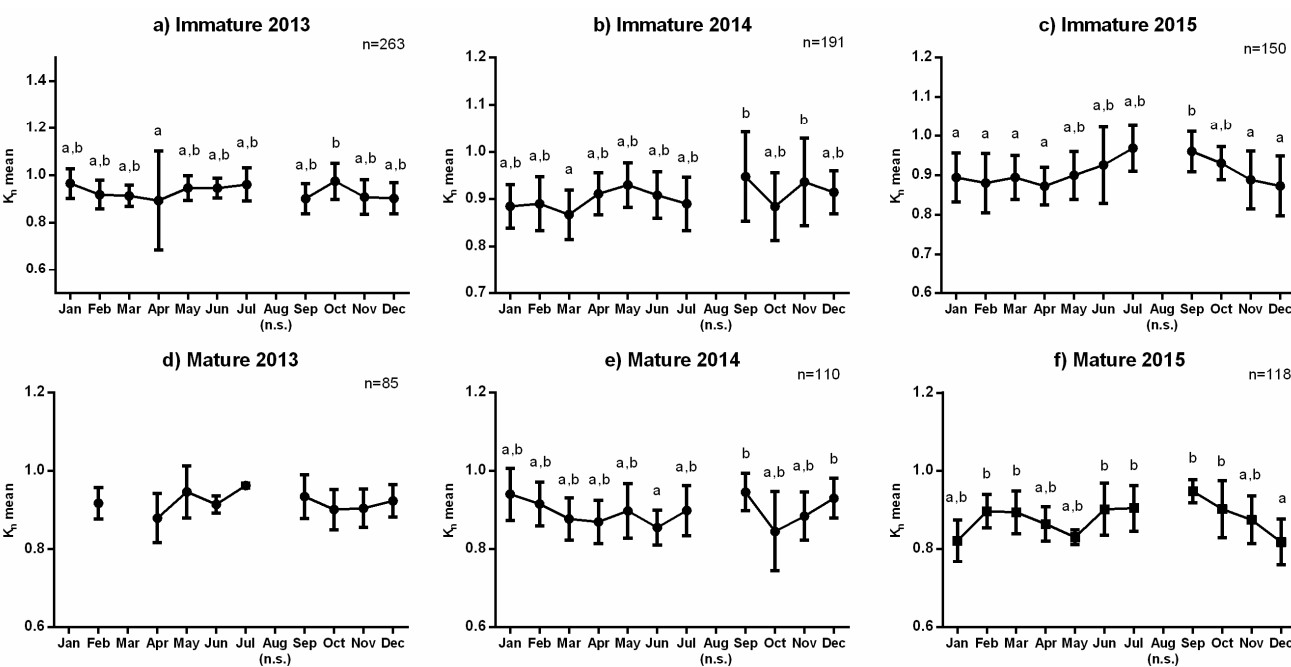

**Figure 4.** Monthly variation of European hake female Le Cren's condition factor ($K_n$) for three years in different ovarian stages. Graphics (**a**–**c**) indicate the females in the immature phase; graphics (**d**–**f**) show females from developing to regenerating phase. In January and March of 2013, mature individuals were not found. In August of the three years, no samples (n.s.) were sampled. The letters indicate a statistically significant difference among groups ($p < 0.05$) determined by Tukey's multiple comparison test.

The analysis of the condition indices per ovarian stage per three years was macroscopically performed (Figure 5a–c). For their histological characteristic, the immature and regenerating stages were grouped. Regarding the GSI, the significantly highest peak was evidenced in the spawning phase for all three years (Tukey's multiple comparison test; $p < 0.05$) (Figure 5a), the trend increased from imm/reg to the spawning stage for all three years but was significant only in 2013 (Tukey's multiple comparison test; $p < 0.05$). For HSI, the tendency of mean values was lowest in the imm/reg stage for all three years (Tukey's multiple comparison test; $p < 0.05$) (Figure 5b). The other mean values were not significantly different among them within the same year, except for 2014, in which the spawning phase showed a significant decrease (Tukey's multiple comparison test; $p < 0.05$) compared to developing and regressing stages. The variation of $K_n$ was less visible compared to the other two indices in the period analyzed (Figure 5c). The only significant difference (Tukey's multiple comparison test; $p < 0.05$) was evidenced in the spawning capable stage compared to the imm/reg and developing stages in 2014 and only compared to imm/reg phase in 2015.

### 3.2. Histological Analysis: Ovarian Classification and Patterns

The female reproductive cycle of the European hake consists of five ovarian reproductive stages according to the histological classification (Table 1). Ovaries from immature individuals exhibited compact ovigerous lamellae with groups of oogonia and previtellogenic oocytes (diameter < 250 µm) and an ovarian wall usually thinner than that of the regenerating stage (Figure 6a). Entry into the developing phase is characterized by the appearance of early and the middle vitellogenic oocytes with a diameter ranged 250–550 µm (Figure 6b). The oocytes take up the yolk proteins. The yolk granules multiply and increase in size, forming a densely packed zone in the inner part of the cytoplasm. The oil droplets proliferate, becoming lipid globules and the zona radiata increases in width. The ovaries from spawning specimens were characterized by the presence of fully grown vitellogenic oocytes (vtg3) and/or migrating nucleus (Mn) stage as well as hydrated oocytes (H) (di-

ameter 550–1150 µm) (Figure 6c–e). Nuclear migration and/or hydration distinguish the actively spawning subphase (Figure 6d). The presence of postovulatory follicles in the same ovary with vtg3, OM, and H was observed (Figure 6e). Extensive atresia and a reduced number of vitellogenic oocytes were considered markers of the regressing stage (Figure 6f). The females in the regenerating phase have spawned at least one time in life. Their ovaries are loose and have thickened wall and CA and LS oocytes are present among previtellogenic oocytes (diameter < 250 µm) (Figure 6g).

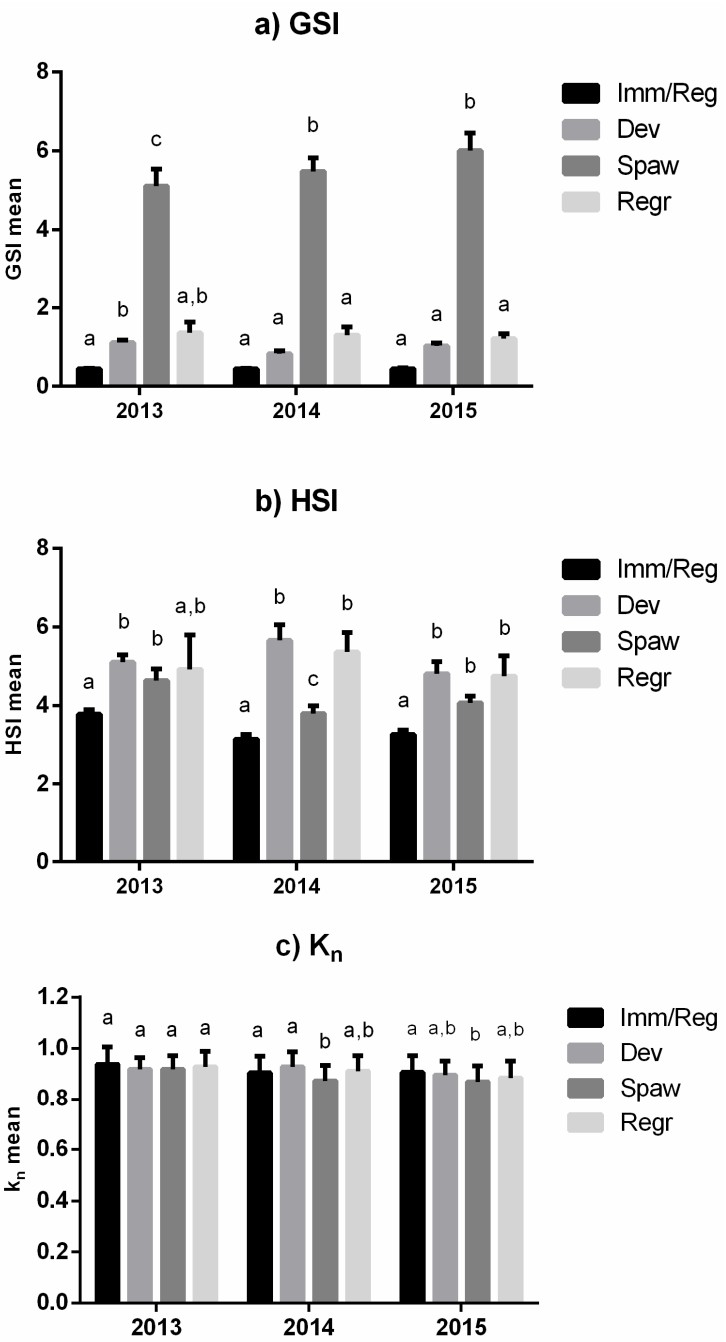

**Figure 5.** Values of the GSI (**a**), HSI (**b**) and $K_n$ (**c**) for females of European hake per ovarian stages for three years. Imm/reg, includes individuals in immature and regenerating stages; dev, developing; spaw, spawning capable; regr, regressing. The letters indicate a statistically significant difference among groups ($p < 0.05$) determined by Tukey's multiple comparison test.

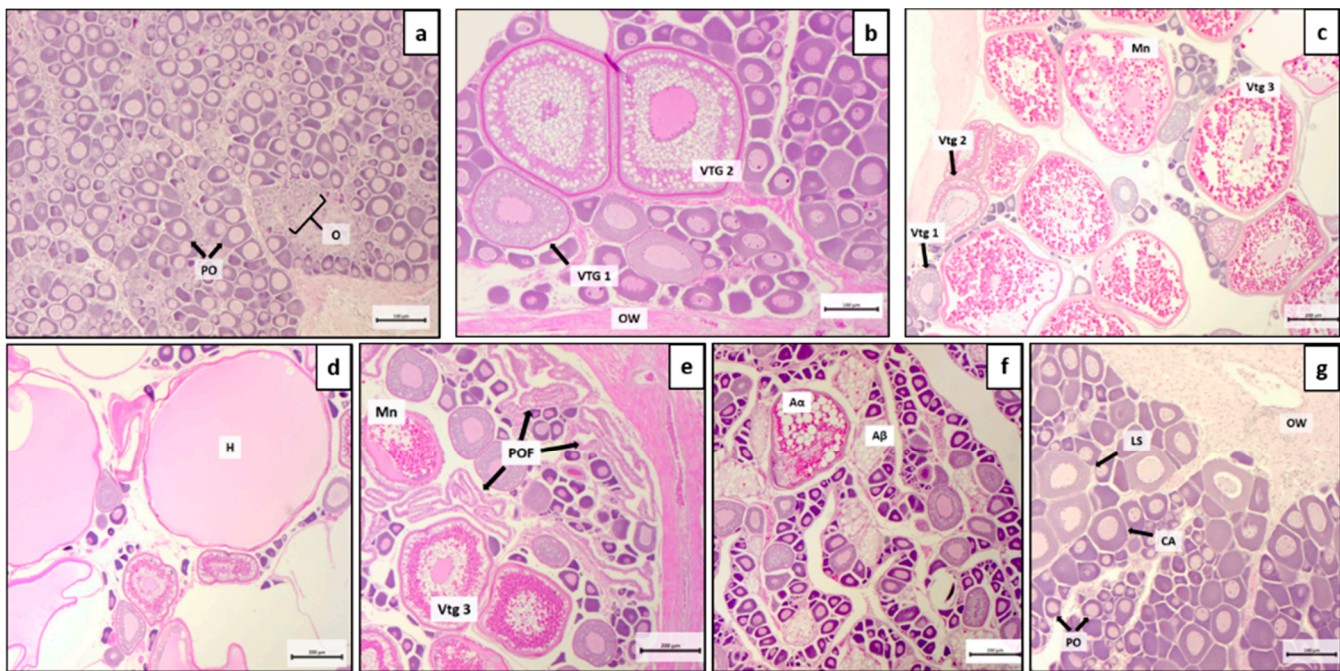

**Figure 6.** Tissue sections of European hake ovaries at different developmental stages. (**a**) Immature; (**b**) developing; (**c**) spawning capable; (**d,e**) actively spawning; (**f**) regressing; (**g**) regenerating. For a,b,g scale bar = 100 μm; for c,d,e,f scale bar = 200 μm. The abbreviations indicate: OW, ovarian wall; O, oogonia; PO, primary oocytes; CA, cortical alveoli; LS, lipid stage; vtg1, vitellogenesis 1; vtg2, vitellogenesis 2; vtg3, vitellogenesis 3; Mn, migrating nucleus; H, hydrated oocytes; POF (postovulatory follicle); Aα, alpha atretic oocyte; Aβ, beta atretic oocyte.

The histological investigation showed a 60.6% similarity with the macroscopic maturity stage classification of the females. Cohen's *k* was 0.45 (95% confidence interval: 0.35–0.55), which corresponds to a "Moderate" level of agreement [36]. Taking into consideration the macroscopic classification, the percentage of agreement between the histological and macroscopic analysis was different among phases. For the imm/reg phase was 91.8%, for the developing stage was 45.8%, for the spawning phase was 85.3%, and for regressing was 2.86%.

### 3.3. Size at First Maturity ($L_{50}$) and Fecundity

$L_{50}$ was estimated macroscopically and histologically to be 30.81 cm and 33.73 cm (Figure 7), respectively.

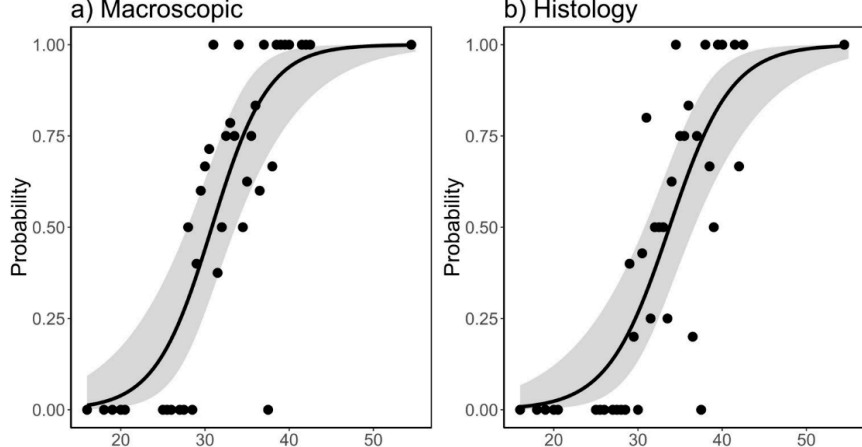

**Figure 7.** Estimated $L_{50}$ and observed the proportion of mature individuals by 1 cm length classes. (**a**) Macroscopic value; (**b**) histological value. The shaded area indicates a 95% confidence interval.

The estimated parameters of the logistic regression were statistically significant ($p < 0.05$) and summarized in Table 2.

**Table 2.** Summary of the L50 estimates based on histological and macroscopic data.

| | Estimate | Std. Error | z value | p-value |
|---|---|---|---|---|
| **Model Estimates HISTOLOGY** | | | | |
| Intercept | −9.17104 | 2.00365 | −4.577 | $p < 0.001$ |
| Total_length_cm | 0.27183 | 0.05925 | 4.588 | $p < 0.0001$ |
| **Model Estimates MACROSCOPIC** | | | | |
| Intercept | −9.15307 | 2.13655 | −4.284 | $p < 0.0001$ |
| Total_length_cm | 0.29703 | 0.06505 | 4.566 | $p < 0.0001$ |

The two curves of $L_{50}$ were not significantly different ($p > 0.05$). The size at which all the females reproduced at least once was 39 cm. The shortest length at which the female specimens entered ovarian development (vitellogenesis) was 26 cm.

Batch fecundity estimation ranged between 7771 (TL 33 cm) and 137,256 (TL 32 cm) hydrated eggs. The shortest total length of a specimen with hydrated eggs in the ovary was 29 cm (batch fecundity of 45,004 eggs) and the longest was 42 cm (batch fecundity 70,519 eggs). The mean value of a relative fecundity was 205 eggs/g of body weight, the lowest value of relative fecundity was 34 eggs/g of body weight found in a female with TL of 33 cm, and the highest value of relative fecundity was 573 eggs/g of body weight derived from a female with TL of 32 cm.

## 4. Discussion

One of the targets of fisheries management is to conserve enough reproductive potential in a population to allow for sustainable exploitation [37]. To achieve this, the reproductive parameters, such as length at first maturity, fecundity, and spawning cycle determination, are usually applied as health indicators of the stock. The present study is the first attempt to highlight an exhaustive knowledge of the reproductive biology of *Merluccius merluccius* in the Northern and Central Adriatic Sea (GSA 17) to complete the scenario of knowledge in the Italian seas.

The easiest, cheapest, and most direct approach to investigate some aspects of reproductive biology is a macroscopic evaluation of the maturity stages of the gonads, but it is not always an accurate method [16,38,39]. This study considered the somatic indices to evaluate the reproductive state of the European hake. We focused on different indices, namely the gonadosomatic index (GSI), the hepatosomatic index (HSI), and Le Cren's condition factor ($K_n$) along three consecutive years. The GSI trend calculated on mature specimens sampled from the GSA17 indicated that the hake reproductive season reached a peak from March to July, even though a presence of spawning females was observed throughout the year, confirming that a protracted spawning period is typical for this species [11,40,41]. Unlike 2014 and 2015, in 2013, a lack of the GSI peak during the reproductive season was observed. The possible reason for this could be related to an exceptional event of dense water that occurred in the winter of 2012 in the Adriatic shelf [42]. Such a phenomenon was characterized by high-velocity currents that caused a cascading event leading to the transport of more than 50% of water volumes from the Northern Basin to the Southern area of the Adriatic Sea [42–44]. Directly or indirectly, unfavorable conditions for the successfulness of the hake reproduction could have emerged, as previously suggested by Recasens et al. [11], in which dense water formation caused intense cascading events shifting the hake reproduction in the Catalan Sea by several months. Thus, the oceanographic events that took place in the Adriatic Sea in 2012 could have negatively impacted the stock of the European hake in 2013 in the area. Regarding the GSI value in different ovarian stages, the

highest mean value was reached at the spawning stage, confirming its correlation with the sexual maturity of gonads.

In the European hake, lipids are stored mainly in the liver, confirming the critical role of this organ for energy storage in this species [19]. The analysis of HSI represents a good proxy of the energy reserve available in fish. In the Central-North Adriatic Sea, HSI showed different trends within the three years studied in both immature and mature females. In 2014 and 2015 for all ovarian phases, the highest recorded values of HSI after summer are probably due to the increase in the food availability for this demersal predator during this period. The energy stored during richer months foodwise is not followed by reproductive activity, allowing the European hake to survive during the winter period, as previously reported in Galician shelf stock [45]. In 2013, the HSI reached the greatest value of the year in the summer months, probably because the absence of the reproductive activity allowed an incessant accumulation of lipids. Based on ovarian developmental stages, the lowest level of HSI was obtained in the imm/reg stage, demonstrating that juvenile fish use lipid energy for growth and do not accumulate them in the liver [46].

Together with HSI, Le Cren's condition factor indicates the energy storage in fishery species. It is given by weight and length data and assumes that heavier fish of a given length are in a better condition [47]. In 2014 and 2015, the $K_n$ showed the highest values in September both for immature and mature individuals, just like HSI. In 2013, the autumn peak shifted one month only for immature females, whereas no variation was shown in mature animals, so we speculated that like for HSI, the $K_n$ also indicates an incessant accumulation of energy in fish during such a period. In the spawning capable phase, the low levels of $K_n$ occurred in 2014 and 2015; only HSI in 2014 could confirm that European hake uses part of the energy accumulated in previous months for reproduction if it is necessary.

The assessment of maturity stages of the ovaries is a crucial step for the estimation of the $L_{50}$, and errors might lead to subsequent biased estimation of the spawning stock biomass [38,39,48]. In this contest, an accurate assessment of sexual maturity is an essential component of effective fisheries management. For European hake, some sexual maturity classifications exist, but often in these studies, a different terminology was used for the oocyte stages or ovarian phases [12,16,28,49]. Furthermore, some authors proposed only an oocyte classification, whereas other authors suggested a macroscopic and histological scale, but without a proper calibration between visual (macroscopic) and histological (microscopic) staging. We used the oocyte stage classification proposed by Candelma et al. [26] and Murua et al. [28] and the standardized terminology for ovarian phases described by Brown-Peterson et al. [27] to obtain a clear and accurate scale on ovarian and oocyte stages validated by the similarity between macroscopic and histological approach. The "Moderate" level of agreement in the assignment of reproductive stages between histological and macroscopic investigation revealed that our macroscopic classification for European hake does not suffer from relatively high error rates. An exception was the regressing phase, whose percentage of agreement was weak. Overall, our scale appeared an accurate tool for the sexual maturity determination by the macroscopic method, notwithstanding some stages require (i.e., the regressing phase and the immature stages) histological analysis to avoid misclassification. Moreover, the histology is the most suitable method for the studies on reproductive biology and it is strictly recommended a regular calibration (over the years) between the two approaches. Histological analysis confirmed that *M. merluccius* is a batch spawner with an asynchronous type of ovarian development.

The $L_{50}$ is a fundamental parameter of fish population dynamics studies and plays a crucial role in species management purposes. In the present study, the length at first maturity was estimated as 30.81 cm by macroscopic evaluation of gonads maturity and 33.73 cm by histological analysis. The $L_{50}$, determined in this work, is comparable to the value of 30.5 cm obtained in a previous study of Zupanovic and Jardas [50] in the same area, while Alegria Hernandez and Jukic [51] estimated an $L_{50}$ equals to 31.3 cm in total length, but both only by macroscopic inspection. In the Southern Adriatic, the size of the

first maturity calculated using the macroscopic method was 31.95 cm [16]. In comparison with previous studies on length at first maturity in the Mediterranean Sea, the values estimated were similar to the measure reported in the present study. On the Egyptian coast, Al-Absawy [14] found an $L_{50}$ of 32.5 cm, while in the Algerian coast [13], it was 30.6 cm. The first maturity size is generally affected by variations of several environmental factors that characterize the different areas, such as the abundance and distribution of local populations, competition for space, and availability of nourishment. Furthermore, the overexploitation of this resource could explain the differences in this parameter that reduces the spawning biomass in different areas [12,16]. The $L_{50}$ of 33.73 cm resulting from histological analysis evidenced that the macroscopic approach underestimates this value, but the two curves were not significantly different, indicating that the histological examination is not strictly necessary.

The $L_{50}$ found in the present study contrasts the decision regulated by Annex III of the Council Regulation (EC) N° 1967/2006, in which the minimum legally allowed fishing size is 20 cm, implying the catch of sexually immature individuals. This excessive fishing effort could lead to a likely decrease in the stock in the area in upcoming years. Accordingly, as reported by Anderson et al. [52], the reduction of the age of a stock and average body size can reduce the ability of exploited species to survive to annual environmental variation.

In the present study, for the first time, batch and relative fecundities in the Northern-Central Adriatic Sea were analyzed. The mean value of a relative fecundity reported in this study (205 eggs/g of body weight) seemed comparable with the previous estimates reported in Catalan waters (204.29 eggs/g of body weight), the North Tyrrhenian (202.35 eggs/g of body weight) [11], and in the Eastern-Central Atlantic (228.33 eggs/g of body weight) [12], yet it was lower than the value estimated in the Central Tyrrhenian and Southern Adriatic Sea (281.6 eggs/g of body weight GSA10 and 262.2 eggs/g of body weight GSA18) [16]. The batch fecundity estimation ranged between 7771 and 137,256 eggs. It was lower compared to the Southern Adriatic Sea and Central Tyrrhenian, probably due to the major dimensions of reproductive animals in these areas compared to Northern and Central Adriatic Sea [16]. In fact, as reported by Recasens et al. [11], El Habouz et al. [12], Carbonara et al. [16] and Korta et al. [53], the relationship between fecundity estimations and hake size influences the total of eggs number per female that increases with size, weight, and gonad weight.

In conclusion, the scenario depicted in the present study evidenced that smaller mature females released fewer eggs per spawn than bigger mature females. As reported by Working Group on Stock Assessment of Demersal Species (WGSAD) [54], the European hake stock of Adriatic Sea results to be overfished and the overexploitation together with a low number of spawned eggs, and unfavorable environmental events could lead to a consequent decrease of the fish resource with the collapse of the stock. To avoid the collapse of European hake, especially in the Central-North Adriatic Sea, continuous monitoring programs, estimation of reproductive potential, and the stock assessment are essential. The easiest, cheapest, and most direct tool to investigate the reproductive stage of European hake, necessary for the determination of reproductive potential, could be represented by the maturity scale provided in this study. Furthermore, the reproductive cycle, the analysis of somatic indices, the $L_{50}$ estimations, and the estimation of fecundity represent valuable information for a scientific decision-making process to establish suitable management measures aimed at tackling the continuing stock decline of European hake in an overfished area as Central-North Adriatic Sea. Finally, the result highlighted in the present study suggested increasing the minimum legal catch size as a tool to preserve this resource in this area.

**Supplementary Materials:** The following are available online at https://www.mdpi.com/article/10.3390/jmse9040389/s1.

**Author Contributions:** M.C. contributed to the acquisition of data, analysis of data, and drafting of the manuscript. S.C. was involved in contributions to conception and design of the manuscript, acquisition of data, and analysis of data and in revising it critically for important intellectual content. L.M. contributed to the analysis and interpretation of data. A.S. contributed to the conception and design of the project and to revising it critically for important intellectual content. D.B., G.R., and G.G. contributed to the analysis of data and to revising it critically for important intellectual content. O.C. conceived the experimental design, contributed to the analysis of data, and revised it critically for important intellectual content. All authors gave the final approval of the version to be published. Each author has participated in the work to take public responsibility for appropriate portions of the content and agreed to be accountable for all aspects of the work in ensuring that questions related to the accuracy or integrity of any part of the work are appropriately investigated and resolved. All authors have read and agreed to the published version of the manuscript.

**Funding:** This work was partially supported by the Italian Ministry of Agricultural, Food, and Forestry Policies (MiPAAF) and European Union (Italian National Programs 2011–2013 and 2014–2016, in the ambit of Data Collection Framework).

**Institutional Review Board Statement:** The sampling procedures did not include any animal experimentation and animal ethics approval was therefore not necessary, under the Italian legislation (D.L. 4 March 2014, n. 26, art. 2).

**Informed Consent Statement:** Not applicable.

**Data Availability Statement:** Data available on request due to restrictions, e.g., privacy or ethical. The data presented in this study are available on request from the corresponding author.

**Acknowledgments:** The authors wish to thank the Filippo Domenichetti and Camilla Croci of Nation-al Research Council (CNR) Institute of Biological Resources and Marine Biotechnologies (IRBIM) Ancona, Italy, and Captain Giordano and crew of the "Orizzonte" vessel for their support in sampling.

**Conflicts of Interest:** Sabrina Colella, Alberto Santojanni, and Oliana Carnevali contributed equally to this work. The authors declare that there is no conflict of interest.

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
