# Peer review of "Aspects of Reproductive Biology of the European Hake (Merluccius merluccius) in the Northern and Central Adriatic Sea (GSA 17-Central Mediterranean Sea)"

_jmse, doi:10.3390/jmse9040389_

Round 1

Reviewer 1 Report

lines: 24-25  is difference in batch fecundity a result of size difference? are size/weight data available online? 
line 51 : possibly unfinished sentence, or the sentence needs rewording
line 94: Were there any selection criteria for this subgroup, or it was randomly selected from all 976 females disregarding size or weight? 
line 124: again were there any selection criteria for these 28 actively spawning females?  
fig 2,3,4 - if possible, add standard deviation for each data point.
262-264: this sentence is confusing and needs rewording
line 335: confusing description of L50 prameter "The length at first maturity (L50)", while at line 151 "The length at which 50% of the females are mature was computed as L50"
line 370: phrase "the most significant dimensions" need rewording or clarification
line 386: probably author means "increasing the minimum legal catch size" instead "increasing the minimum legal catching"

Author Response

Response to Reviewer 1 Comments

POINT 1: lines 24-25: is difference in batch fecundity a result of size difference? are size/weight data available online? 

Response 1: Reporting Recasens et al. (2008), Korta et al. (2010) El Habouz et al. (2011), and Carbonara et al. (2019) “The relationship between fecundity estimations and hake size influences the number of eggs per female that increases with size, weight, and gonad weight.”  This sentence has been inserted in lines 431-434 of the manuscript. This reference was added “Korta, M.; Dominguez-Petit, R.; Murua, H.; Saborido-Rey, F. Regional variability in reproductive traits of European hake Merluccius merluccius L. populations. Fish. Res. 2010, 104, 64–72, doi:10.1016/j.fishres.2009.03.007.”

POINT 2: line 51: possibly unfinished sentence or the sentence needs rewording

Response 2: line 51-: the new sentence was introduced: “In this regard, female producing eggs, contributes with a greater extent of male to embryonic development success for this the acquisition of information on reproductive cycle of the female Adriatic stock, including length at first maturity, fecundity, and spawning cycle is essential to quantify the reproductive potential of this population”

POINT 3: line 94: Were there any selection criteria for this subgroup, or it was randomly selected from all 976 females disregarding size or weight? 

Response 3: “The female specimens at different maturity stages were randomly chosen but choosing a group that was representative for all sizes.” It was added to the manuscript at line 101.

POINT 4: line 124: again were there any selection criteria for these 28 actively spawning females?  

Response 4: 28 females in the advanced maturity stage and showing the absence of recent post-ovulatory follicles in the histological analysis represents the total number of females available. It was added to the manuscript at line 130

POINT 5: fig 2,3,4 - if possible, add standard deviation for each data point.

Response 5: modified as requested

POINT 6: 262-264: this sentence is confusing and needs rewording

Response 6: Line 313-315: The sentence was rephrased in “the lowest value of relative fecundity was 34 eggs/g of body weight found in a female with TL of 33 cm, and the highest value of relative fecundity was 573 eggs/g of body weight derived from a female with TL of 32 cm.”

POINT 7: line 335: confusing description of L50 parameter "The length at first maturity (L50)", while at line 151 "The length at which 50% of the females are mature was computed as L50"

Response 7: Line 399: This is correct in “The L50 is a fundamental parameter”.

POINT 8: line 370: the phrase "the most significant dimensions" need rewording or clarification

Response 8: The sentence was revised “It was lower compared to the Southern Adriatic Sea and Central Tyrrhenian, probably due to the major dimensions of reproductive animals in these areas compared to Northern and Central Adriatic Sea”.

POINT 9: line 386: probably author means "increasing the minimum legal catch size" instead of "increasing the minimum legal catching"

Response 9:  the sentence was modified according to the suggestion

Reviewer 2 Report

This study describes the annual variation in the mature length and maturation state of females of the European Hake (Merluccius merluccius), and contains important data on the actual reproduction of the fish. However, the structure of the paper is not appropriate, excessive statistical processing is used, and there is a lack of explanation regarding the reliability of the data, so I judge that the paper is not ready for acceptance at this stage.

Main points

  1. The authors should clearly state in the preface why they chose to study only female individual.
  2. There is a possibility that the data included in Figs. 1-2 include data on immature fish. Therefore, the life year cycles of adult and immature fishes are discussed together. The fact that the life cycle of many fishes differs between adults and immatures has not been taken into account. In my opinion, this paper should first present the results of the analysis of mature body length, and then present the results of the analysis of HSI and Kn data separately for adult and immature fish.
  3. Statistical processing is excessive. Usually, for a single data set, one should choose between one ANOVA or one multiple comparison as the overall statistical test. If a significant difference occurs somewhere in the multiple comparisons, it is obvious that there is a significant difference in the overall test. It is not necessary to write both.

Minor points

Why are the ovarian developmental stages in L104-105 and Table 1 different from those in Fig.1? (There is no regenerating in Fig.1). It would be better to standardize one of them, because readers will be confused.

Figs.1-2: It would be better to insert “August” with the label "No data". Also, the sample size "n=" should be added to each month's data, otherwise the reliability of the data cannot be known. The vertical axis should be labeled as to what it represents.

Figs.1-2: Shouldn't the data be composed of only those individuals whose length is greater than or equal to that of a fully adult fish?

Fig.2 August without data should also be added to the X-axis. also related to Fig.1, if the data include immature fish, it may not be appropriate to use mean only because the GSI data for each month are not normally distributed (probably bimodal). It would be clearer to show the data for each month in a box-and-whisker diagram.

Table 2: Consider carefully the measurement precision of the estimates given in Table 2. To indicate a value with six significant digits, at least n>1000000 is required. Also, when the value of p is calculated as 0.000005, etc., the notation p<0.001 or p<0.0001 is sufficient. Differences between values smaller than these values are statistically almost meaningless.

L285-287 Cause and effect are reversed.

L307-312: There is no direct evidence in this study that food availability affects the proportion of individuals that reach maturity, so this is not a scientific argument. It would be better if evidence could be provided, but if this is not possible, at least collateral data sufficient to guide the discussion should be cited.

Author Response

Please, see the manuscript in the attachment.

Response to Reviewer 2 Comments

Main points

Point 1: The authors should clearly state in the preface why they chose to study only female individual.

Response 1:  Line 51-54 the suggestion was added: “In this regard, female producing eggs, contributes with a greater extent of male to embryonic development success for this the acquisition of information on reproductive cycle of the female Adriatic stock, including length at first maturity, fecundity, and spawning cycle is essential to quantify the reproductive potential of this population”

Point 2: There is a possibility that the data included in Figs. 1-2 include data on immature fish. Therefore, the life year cycles of adult and immature fishes are discussed together. The fact that the life cycle of many fishes differs between adults and immatures has not been taken into account. In my opinion, this paper should first present the results of the analysis of mature body length, and then present the results of the analysis of HSI and Kn data separately for adult and immature fish.

Response 2: Figure 1 is only the percentage of females in different maturity stages, so the immature are included. But, as suggested by the reviewer, the analysis of GSI, HSI, and Kn has been conducted again following the suggestion raised. Consequently, all images, the comments in the Materials&Methods, Results, and Discussion were changed along with the text.

Point 3: Statistical processing is excessive. Usually, for a single data set, one should choose between one ANOVA or one multiple comparison as the overall statistical test. If a significant difference occurs somewhere in the multiple comparisons, it is obvious that there is a significant difference in the overall test. It is not necessary to write both.

Response 3: The statistical terminology was incorrect; it was changed in lines 140 and 212.

Minor points

Point 4: Why are the ovarian developmental stages in L104-105 and Table 1 different from those in Fig.1? (There is no regenerating in Fig.1). It would be better to standardize one of them, because readers will be confused.

Response 4: correct.

Point 5: Figs.1-2: It would be better to insert “August” with the label "No data". Also, the sample size "n=" should be added to each month's data, otherwise the reliability of the data cannot be known. The vertical axis should be labeled as to what it represents.

Response 5: August has been included. In Figure 2 is reported the total number of the animal per year. The sample size is reported in a supplementary table attached at the end of the manuscript for revision time.

Point 6: Figs.1-2: Shouldn't the data be composed of only those individuals whose length is greater than or equal to that of a fully adult fish?

Response 6: As reported in point 2, all suggestions were applied.

Point 7: Fig.2 August without data should also be added to the X-axis. also related to Fig.1, if the data include immature fish, it may not be appropriate to use mean only because the GSI data for each month are not normally distributed (probably bimodal). It would be clearer to show the data for each month in a box-and-whisker diagram.

Response 7: As reported in point 2, all suggestions were included.

Point 8: Table 2: Consider carefully the measurement precision of the estimates given in Table 2. To indicate a value with six significant digits, at least n>1000000 is required. Also, when the value of p is calculated as 0.000005, etc., the notation p<0.001 or p<0.0001 is sufficient. Differences between values smaller than these values are statistically almost meaningless.

Response 8: the table was upgraded.

Point 9: L285-287 Cause and effect are reversed.

Response 9: changed in “in which dense water formation caused intense cascading events shifting the hake reproduction in the Catalan Sea by several months.”

Point 10: L307-312: There is no direct evidence in this study that food availability affects the proportion of individuals that reach maturity, so this is not a scientific argument. It would be better if evidence could be provided, but if this is not possible, at least collateral data sufficient to guide the discussion should be cited.

Response 10: Discussion was modified according to Ref 2 suggestions 

Round 2

Reviewer 2 Report

The present study has been improved to analyse adult and immature fish separately, which has enabled a correct assessment of the relationship between maturity status and nutritional status.

In response to my remarks on the inadequacy of the statistical treatment, the authors attempted to improve it and to unify it into a one-way ANOVA. However, they did not seem to have a sufficient understanding of the use of different statistical methods, and as a result there are many mistakes in the text and figures of the paper. The reason why I have stated that it is not appropriate to apply two different statistical tests to one data set at the same time is that it increases “the problem of multiplicity”. If we perform two statistical tests on the same data at the level of p=0.05, the probability of a significant difference in one of the tests increases to p=0.10. Therefore, for any statistical method, it is better to reduce the number of statistical tests.

  1. If the authors had chosen a one-way analysis of variance, they would have performed only one test on each data set, and if it was significant, they would not have performed any further statistical treatment, but would have simply stated in which month the peak was found. In this case, all the results of the multiple comparisons should be eliminated, such as where the significant differences are found, such as a, b and c in the figure. The "*" with the line is also unnecessary, and only one p-value should be shown for each panel.

For example, the description of L173-181 is as follows

“In figure 2 is represented the GSI trend of mature females during three years of sampling. The GSI values were lowest in 2013 than the other two years. In 2013, the highest value was recorded in April (one-way ANOVA; p<0.05) (Fig. 2a). In 2014 and in 2015 the highest peak was observed in June and the lowest in September (Fig. 2b-c) (both p<0.05). Like June, also July showed the highest peak in 2014.” [L176-177: “(Fig 2b-c)” is “(Fig. 2b-c)”.]

  1. On the other hand, if the authors chose the multiple comparisons method, they should explain in writing which months are significantly higher or lower than which months, and at the same time, they should write only a, b, c in the figures. The "*" in the line should never be added.

For example, the description of L173-181 is as follows

“In Figure 2 is represented the GSI trend of mature females during three years of sampling. The GSI values were lowest in 2013 than the other two years. In 2013, the highest value was recorded in April with a significant difference only compared to October (Tukey's multiple comparison test; p<0.05) (Fig. 2a). In 2014, the GSI in June was significantly different compared to February, September, October, November, and December (Fig. 2b), and that in July was significantly different to September, December, and February. In 2015, June was significantly different only compared to September (p<0.05) (Fig. 2c).”

Either method, 1 or 2, is acceptable, but in my opinion it is not necessary to describe at length the detailed statistical differences. What is important is that the statistical tests used are consistent throughout the paper. I will not write more detailed corrections, but the text and the figures should be carefully adjusted to each statistical test.
